# Segment as You Wish: Free-Form Language-Based Segmentation for Medical Images

## Abstract

Medical imaging is crucial for diagnosing a patient's health condition, and accurate segmentation of these images is essential for isolating regions of interest to ensure precise diagnosis and treatment planning. Existing methods primarily rely on bounding boxes or point-based prompts, while few have explored text-related prompts, despite clinicians often describing their observations and instructions in natural language. To address this gap, we first propose a RAG-based free-form text prompt generator, that leverages the domain corpus to generate diverse and realistic descriptions. Then, we introduce `FLanS`, a novel medical image segmentation model that handles various free-form text prompts, including professional anatomy-informed queries, anatomy-agnostic position-driven queries, and anatomy-agnostic size-driven queries. Additionally, our model also incorporates a symmetry-aware canonicalization module to ensure consistent, accurate segmentations across varying scan orientations and reduce confusion between the anatomical position of an organ and its appearance in the scan. `FLanS` is trained on a large-scale dataset of over 100k medical images from 7 public datasets. Comprehensive experiments demonstrate the model's superior language understanding and segmentation precision, along with a deep comprehension of the relationship between them, outperforming SOTA baselines on both in-domain and out-of-domain datasets.

## 1 Introduction

Medical imaging is crucial in healthcare, providing clinicians with the ability to visualize and assess anatomical structures for both diagnosis and treatment. Organ segmentation is vital for numerous clinical applications, including surgical planning and disease progression monitoring (Wang et al., 2022b; Du et al., 2020; Shamshad et al., 2023). However, accurately segmenting organs and tissues from these medical images, i.e., medical image segmentation (MIS), remains a significant challenge due to the variability in patient positioning, imaging techniques, and anatomical structures (Pham et al., 2000; Xiao & Sun, 2021). Recent advancements in large foundation models, such as Segment Anything Model (SAM) (Kirillov et al., 2023) and MedSAM (Wu et al., 2023), have shown promise in achieving more accurate and faster MIS. These models often require the users to input a predefined category name, a box, or a point as a prompt. However, in real-world scenarios, clinicians often rely on natural language commands to interact with medical images, such as "*Highlight the right kidney*" or "*Segment the largest organ*". An accurate segmentation model with flexible text comprehension capability is therefore essential for a wide range of clinical applications.

**The first challenge** lies in the development of a segmentation model that can handle text prompts (Zhao et al., 2024), offering greater flexibility and adaptability in real-world clinical environments. Unlike traditional models that rely on bounding boxes (Bboxes) or point prompts, this method should allow clinicians to use *free-form* natural language commands and streamline the diagnostic process by enabling intuitive, verbal interactions. For *free-form* text, we provide two conceptual definitions as follows: (1) *Anatomy-Informed Segmentation*, where the user has explicit knowledge of the organ or relevant pathology to be segmented; (2) *Anatomy-Agnostic Segmentation*, where the user lacks medical knowledge about a specific organ or CT scan and hence queries based on positional information, organ sizes or other visible characteristics. This scenario is more common for individuals such as students or patients without formal medical training. An exemplar illustration is shown below [1]:

---

[1] All of the images in this paper are best view in color.

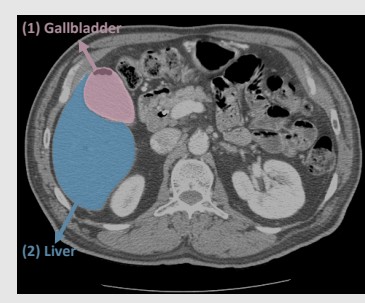

**Example (1) : Anatomy-Informed Segmentation:** *An abdominal CT scan is recommended to evaluate for the presence of gallstones or any fluid accumulation around the gallbladder. —* *This professional diagnosis snippet indicates the most cared segmentation area is:* ***Gallbladder***.

**Example (2): Anatomy-Agnostic Segmentation:** *I would love to have the leftmost organ segmented in this CT scan. —* *This description is agnostic to the medical name of the organ, but it indicates the segmentation target from positional semantics:* ***Liver***.

To learn a free-form text-supportive MIS model, text prompt generation towards the groundtruth mask is a primary step. Instead of using labor-intensive manual labeling to match with the masks, we propose a retrieval augmented generation (RAG) fashion (Lewis et al., 2020) method that automates text query generation using corpus embeddings collected from three resources (clinical expert records, non-expert queries, and synthetic queries). This approach guarantees that the generated query prompts capture various forms of language use across different demographic groups. Based on the text queries, we propose **FLanS**, a **f**ree-form **lan**guage-based **s**egmentation model that can accurately interpret and respond to *free-form* prompts either professional or straightforward, ensuring accurate segmentation across a variety of query scenarios.

**Another challenge** in text-based medical imaging segmentation arises from the variability in scan orientation. Factors such as patient positioning (e.g., supine vs. prone), different imaging planes (axial, coronal, sagittal), reconstruction algorithms and settings, and the use of portable imaging devices in emergency settings can cause organs to appear in unexpected locations or orientations. The scan orientations even differ between well-preprocessed datasets, such as AbdomenCT-1K (Ma et al., 2022) and BTCV (Gibson et al., 2018), as shown in Fig. 1. This variability can confuse segmentation models, making it difficult to distinguish between the anatomical position of an organ and its appearance in a scan. For instance, the right kidney may appear either on the left or the right side of a rotated scan, leading to inaccurate segmentations. To address this challenge, we integrate the symmetry-aware canonicalization module as a crucial step in our model architecture (Kaba et al., 2022; Mondal et al., 2023), which ensures the model produces consistent segmentations regardless of the scan's orientation, enhancing its accuracy across diverse medical images (Cohen & Welling, 2016; Weiler & Cesa, 2019). Additionally, incorporating symmetry improves sample efficiency and generalizability, which is well-suited for medical imaging tasks where labeled datasets are limited (Wang et al., 2022a; 2021; Zhu et al., 2022; Thomas et al., 2018).

Our key contributions in this paper are summarized as follows:

- We employ RAG techniques for free-form text prompt generation for various anatomical structures containing diverse anatomy-informed and anatomy-agnostic queries. Stems from the vectorized embedding of clinical reports, produced query data employs the realistic tones and word usage.
- We present a novel medical image segmentation model, **FLanS**, that exhibits a deep understanding of the relationship between text descriptions and medical images. It uniquely supports free-form text segmentation and employs a symmetry-aware canonicalization module to handle variability in scan orientation, as in Table. 1.
- Our model training uses ~100k medical images from 7 public datasets, covering 24 organs, along with diverse text prompts. This ensures the model generalizes across diverse anatomical structures and clinical scenarios and can be easily extended to new organs with upcoming datasets.
- We demonstrate the **FLanS**'s effectiveness on both in-domain and out-of-domain datasets, and perform ablation studies to validate the contributions of each component in our model design.

## 2 RELATED WORK

**Medical Image Segmentation**   Medical image segmentation (MIS) aims at accurately delineating anatomical structures in medical images. Traditionally, MIS methods tend to segment the correct regions from an image that accurately reflects the input query (Azad et al., 2024). The researchers improve the performance of MIS methods by either optimizing segmentation network design for improving feature representations (Chen et al., 2018b; Zhao et al., 2017; Chen et al., 2018a; Gu et al., 2020), or improving optimization strategies, e.g., proposing better loss functions to address class

Table 1: FLanS uniquely supports all prompt types, including free-form text, and is symmetry-aware.

| Model | Prompt Type | | | | Symmetry Aware |
|---|---|---|---|---|---|
| | Label | Point | Bbox | Text | |
| SAM-U (Deng et al., 2023) | ✗ | ✔ | ✔ | ✗ | ✗ |
| SAMed (Zhang & Liu, 2023) | ✗ | ✔ | ✔ | ✗ | ✗ |
| AutoSAM (Hu et al., 2023) | ✗ | ✔ | ✔ | ✗ | ✗ |
| MedSAM (Ma et al., 2024) | ✗ | ✔ | ✔ | ✗ | ✗ |
| MSA (Wu et al., 2023) | ✗ | ✔ | ✔ | ✗ | ✗ |
| Universal (Liu et al., 2023b) | ✔ | ✗ | ✗ | ✗ | ✗ |
| **FLanS (ours)** | ✔ | ✔ | ✔ | ✔ | ✔ |

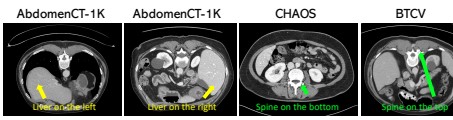

Figure 1: Example CT images from different datasets show significant variations in orientation, which highlight the need for a symmetry-aware (equivariant) model to ensure consistent segmentation performance across diverse scan orientations.

imbalance or refining uncertain pixels from high-frequency regions to improve the segmentation quality (Xue et al., 2020; Shi et al., 2021; You et al., 2022). However, they require a pre-known medical region from the user as an input for segmentation on where it is expected to be segmented and a precise match between the segment's name and the labels used in the training set, restricting their flexibility in real-world application. Another category of methods are SAM-based approaches (Kirillov et al., 2023; Ma et al., 2024; Zhu et al., 2024) that mainly rely on the Bboxes or points as prompts for segmentation. While such methods do not need strict labels, they neglect the descriptive understanding of the image, revealing a deficiency in performing arbitrary description-based segmentation, in comparison, our method handles well in *Labels*, free-form *Text* prompts without losing ability of *Point* and *Bbox*, as shown in the Table. 1.

**Text Prompt Segmentation**   Text prompt segmentation, also referred to as expression segmentation (Hu et al., 2016), utilizes natural language expressions as input prompts for image segmentation tasks Liu et al. (2023a), moving beyond the traditional reliance on class label annotations (Liu et al., 2021), such as nn-Unet (Isensee et al., 2018), and Swin-unet (Cao et al., 2022). Early research in this area employed CNNs and RNNs for visual and textual feature extraction, which were later combined through feature fusion for segmentation (Li et al., 2018). The success of attention mechanisms further inspired a new line of work (Shi et al., 2018; Ye et al., 2019). More recently, transformer-based architectures have improved segmentation performance by using either carefully designed encoder-based feature fusion modules (Feng et al., 2021; Yang et al., 2022; Kim et al., 2022) or decoder-based approaches (Wang et al., 2022c; Lüddecke & Ecker, 2022; Ding et al., 2021). Among these, (Zhou et al., 2023) introduced a text-promptable mask decoder for efficient surgical instrument segmentation. However, no existing work has focused on free-form language segmentation for diagnosis-related medical imaging tasks as introduced in this work.

**Equivariant Medical Imaging**   Equivariant neural networks ensure that their features maintain specific transformation characteristics when the input undergoes transformations, and they have achieved significant success in various image processing tasks (Cohen & Welling, 2016; Weiler & Cesa, 2019; Cohen et al., 2019; Bronstein et al., 2021). Recently, equivariant networks have also been applied to medical imaging tasks, including classification (Winkels & Cohen, 2018), segmentation (Kuipers & Bekkers, 2023; Elaldi et al., 2024; He et al., 2021), reconstruction (Chen et al., 2021), and registration (Billot et al., 2024). Equivariance can be incorporated in different ways, such as through parameter sharing (Finzi et al., 2021), canonicalization (Kaba et al., 2022), and frame averaging (Puny et al., 2021). In our work, since we leverage a pretrained segmentation network, we achieve equivariance/invariance through canonicalization (Mondal et al., 2023), which, unlike other methods, does not impose architectural constraints on the prediction network.

## 3 METHODOLOGY

In this section, we introduce a paradigm to equip the segmentation model with free-form language understanding ability while maintaining high segmentation accuracy. It employs the RAG framework to generate text prompts based on real world clinical diagnosis records. The generated free-form queries, anchored on the corresponding organ labels, are used to train a text encoder capable of efficiently interpreting the segmentation intentions (e.g., different interested organs disclosed in anatomy-informed or anatomy-agnostic prompts) and guiding the segmentation network. We also incorporate a canonicalization module, which can transform input images with arbitrary orientations into a learned canonical frame, allowing the model to produce consistent predictions regardless of the input image orientation.

**Preliminaries of SAM Architecture** SAM (Kirillov et al., 2023) contains three main parts: (1) an image encoder that transforms images into image embeddings; (2) a prompt encoder that generates prompt embeddings; (3) a mask decoder that outputs the expected segmentation mask based on the image and prompt embeddings. Given a corresponding input medical image $x \in \mathcal{X}$ and a relevant prompt $p \in \mathcal{P}_x$. The image encoder embeds $x$ into $z_x$ that $z_x = \text{Encoder}^{\mathcal{X}}(x)$, similarly the prompt embedding $z_p = \text{Encoder}^{\mathcal{P}}(p)$. The mask decoder predicts the segmentation result (mask) by $\hat{m}_x^p = \text{Decoder}(z_x, z_p)$. While the SAM model provides $\text{Encoder}^{\mathcal{P}}$ for spatial prompts (e.g. Bbox or point), the integration of text-based prompts has been less explored. In text-based medical images segmentation, natural language prompts require specialized learning to effectively capture clinical terminology and segmentation intent.

### 3.1 THE RETRIEVAL AUGMENTED QUERY GENERATOR

**Anatomy-Informed Query** To equip a MIS model $\mathcal{M}$ with language comprehension abilities, it is essential to prepare a suitable natural language query [2] corpus $\mathcal{C}$ in correspondence with the target organ label set $\mathcal{L} = \{l_1, l_2, ...l_n\}$, where $l_1 = $ *Liver*, $l_2 = $ *Kidney*, etc., as in Appendix Fig. 9. Since manual annotation is time-consuming and can be biased towards individual linguistic habits, we designed a RAG-based free-form text prompts generator to automate this process. RAG allows pre-trained LLMs to retain their free-form language generation capabilities while incorporating domain-specific knowledge and style from the provided data source $\mathcal{S}$. We collect corpus from three types of data sources. Two of these, $\mathcal{S}_1 = $ *Domain Expert*, $\mathcal{S}_2 = $ *Non-Expert*, serve as the corpus set to simulate various styles of descriptions for segmentation purposes,. The third source, $\mathcal{S}_3 = $ *Synthetic*, is directly generated by GPT-4o to imitate descriptions for segmentation purposes.

For $\mathcal{S}_1 = $ *Domain Expert*, we collected over 7,000 reports written by doctors and identified 4,990 clinical diagnosis records that are relevant to 24 labeled organs for this study. After de-identification, we embed such Electronic Medical Records (EMRs) into semantic vector space through Med-BERT (Rasmy et al., 2021), which outperforms the general language embedding models such as Bert or GPTs in the bioinformatics context understandings. Then, we built a retrieval augmented generation fashion generator agent **G**, as shown in Fig. 2, provided with medical domain corpus and practitioner's language usage preference. It retains the original LLM's natural language ability such as sentences extension and rephrasing. Finally, we construct a

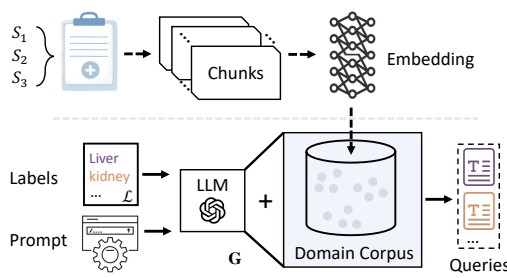

Figure 2: The RAG Free-form Query Generator. The domain corpus, from the EMRs embedding, completes the retrieval augmentation and enhances the LLMs with the clinical way of query.

query prompt template: "***System:** You are an agent able to query for segmenting label* {*Liver*} *in this* {*CT*} *scan. Please write the query sentence and output it.*" Given a label $l_i = $ *Liver*, where $l \in \mathcal{L}$ regarding an arbitrary organ label with *CT* modality, the **G** produces a free-form query $q_l^i$, this query is taken as prompt in the later text-aware segment model training. E.g., "*(1) Examine this CT scan to determine the extent of hepatic damage present. (2) As the symptoms suggest cirrhosis, we should analyze the related part in this CT scan for any signs of the disease*". These retrieved augmented results show that the interested organ may not always be explicitly mentioned, but can be inferred based on terms like 'cirrhosis' and 'hepatic', which are all liver-specific illnesses in clinical practice.

For $\mathcal{S}_2 = $ *Non-Expert*, we collected queries from people without medical training who lack knowledge of the anatomy structures to formulate the segmentation queries. For $\mathcal{S}_3 = $ *Synthetic*, the corpus is directly generated by LLMs. Both $\mathcal{S}_2$ and $\mathcal{S}_3$ are combined with $\mathcal{S}_1$ and processed by **G** to produce diverse and rich expression text queries for any given organ.

**Anatomy-Agnostic Query** Anatomy-agnostic queries are crucial for training models to handle more plain descriptions (i.e., positions, sizes) that lack explicit organ names or related anatomy information. To align the anatomy-agnostic queries, $\mathcal{Q}$, with training images and their ground truth masks, we follow the process shown in Fig. 3. Given a training sample $x$, we first retrieve

---

[2]Throughout the paper, we use the terms "query" and "prompt" interchangeably.

spatial information for each of its mask $m_x^{(i)}$ using Bboxes, deriving spatial categories based on their positions and sizes, $k \in \mathcal{K}$, where the set $\mathcal{K} = \{k^{1*}, k^{2*}, \ldots, k^{6*}\}$ represents six categories: largest, smallest, left-most, right-most, upmost, and bottom. The RAG generator $\mathbf{G}$ then extends this information into full language descriptions for the masks that belong to one of these six categories, generating anatomy-agnostic text queries to augment $\mathcal{P}_x$ for each $x \in \mathcal{X}$. This pipeline, as Fig. 3, ensures sufficient anatomy-agnostic queries are provided to train the model to segment the accurate organ masks without needing to know the organ label names.

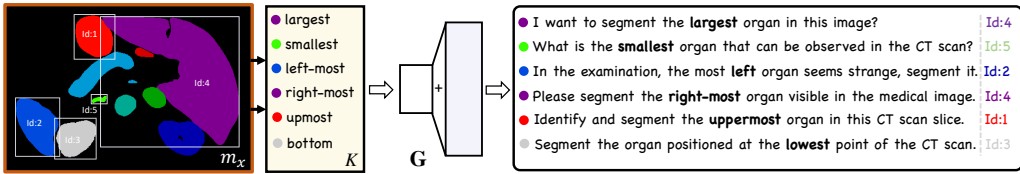

Figure 3: Spatial features extracted from the Bboxes of ground truth masks are processed by the RAG query generator $\mathbf{G}$ to produce anatomy-agnostic queries.

## 3.2 FREE-FORM LANGUAGE SEGMENTATION FOR MEDICAL IMAGES

After generating a large corpus of free-form text queries via our retrieval augmented query generator, the next step is to align these queries with medical imaging segmentaion tasks.

**Anatomy-Informed Segmentation**    For free-form anatomy-informed text prompts, the text encoder must learn embeddings that group similar organ segmentation intents together while clearly separating unrelated intents in distinct semantic clusters. We adopt the CLIP (Radford et al., 2021) as the foundation of text encoder for its capability of understanding semantics. Given a text prompt $p \in \mathcal{P}_x$ associated with the image $x$, the CLIP text encoder converts it into an embedding vector $\mathbf{t}_p$ in a shared embedding space: $\mathbf{t}_p = \text{Encoder}^{\mathcal{P}}(p) \in \mathbb{R}^D$, where $D$ is the dimensionality of the text embedding space. To further strengthen the model's ability to differentiate between organ segmentation, we introduce an intention head on top of the text embeddings by CLIP. This head is a linear layer $\mathbf{W}_{\text{cls}} \in \mathbb{R}^{C \times D}$, where $C = 24$ is the number of organ class. The intention logits $\mathbf{y}_p$ are derived for each encoded vector $\mathbf{t}_p$: $\mathbf{y}_p = \mathbf{W}_{\text{cls}}\mathbf{t}_p + \mathbf{b}_{\text{cls}}$. Given a corresponding medical image embedding $z^x$, we train the model by following loss function:

$$L = \underset{\{\mathbf{W}_{\text{cls}}, \mathbf{b}_{\text{cls}}, \mathbf{W}^E, \mathbf{W}^D, \mathbf{W}^P\}}{\arg\min} \frac{1}{|\mathcal{X}|} \sum_{x \in \mathcal{X}} \frac{1}{|\mathcal{P}_x|} \sum_{p \in \mathcal{P}_x} [\mathcal{L}_{\text{Dice}}(\hat{m}_x^p, m_x^p) + \mathcal{L}_{\text{ce}}(\hat{m}_x^p, m_x^p) + \mathcal{L}_{\text{ce}}(\mathbf{y}_p, l_p)] \quad (1)$$

where $\hat{m}_x^p = \text{Decoder}(z_x, \mathbf{t}_p)$ and $m_x^p$ are predicted and ground truth masks. $l_p \in [0, ..., 23]$ is the ground truth organ class for the prompt. $\mathbf{W}^E$, $\mathbf{W}^D$ and $\mathbf{W}^P$ represent the image encoder, decoder and CLIP text encoder weights, respectively. We use both Dice loss $\mathcal{L}_{\text{Dice}}$ and cross-entropy loss $\mathcal{L}_{\text{ce}}$ for predicted masks. The classification loss $\mathcal{L}_{\text{ce}}(\mathbf{y}_p, l_p)$ encourages the model to correctly classify organs based on text prompts, ensuring the text embedding aligns with the intended organ class.

**Anatomy-Agnostic Segmentation**    For anatomy-agnostic descriptions, which do not explicitly mention specific organs but instead focus on spatial attributes (e.g., "leftmost", "largest"), the model must learn from spatial features $k_x \in \mathcal{K}$ to pair with the corresponding mask $m_x^k$ for every $x \in \mathcal{X}$. Anatomy-agnostic queries share the same embedding space as anatomy-informed queries, but $k_x$ is not necessarily associated with a specific organ. In this case, we use the same loss function as shown in Eq. 1 but without the last classification term.

## 3.3 SEMANTICS-AWARE CANONICALIZATION LEARNING

We incorporate roto-reflection symmetry (Cohen & Welling, 2016) into our architecture for two key reasons: 1) Organs and anatomical structures can appear in various orientations and positions due to differences in patient positioning, imaging techniques, or inherent anatomical variations. Equivariance ensures that the model's segmentation adapts predictably to transformations of the input image. 2) We aim to ensure our model reliably interprets and segments organs that have positional terms in their names, such as "left" or "right kidney" from text prompts regardless of the scan's orientation, thereby enhancing the model's robustness and accuracy.

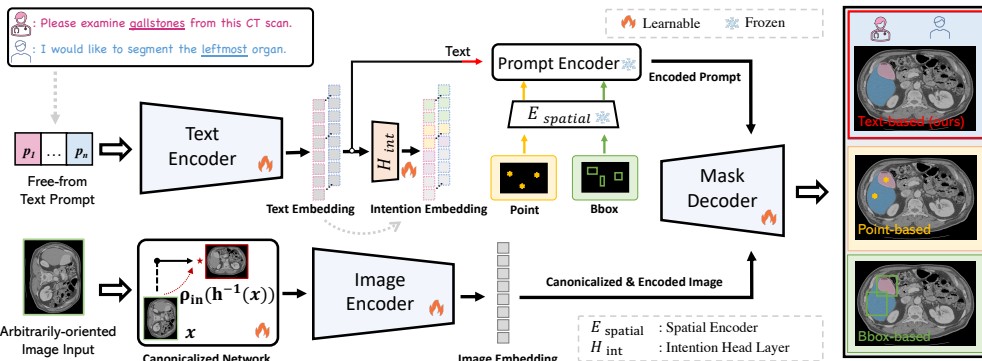

Figure 4: The architecture of our proposed model **FLanS**. First, given a set of free-form text prompts $p_1..., p_n$, the text encoder gets the text embedding, and then passes through the learned *Intention Head Layer* that maps the embedding to a space with explicit intention probabilities, which is useful for the **FLanS** model weight updating as in Eq. 1. Second, we have trained a *Canonicalized Network* that transforms any medical image with arbitrary orientation into a canonicalization space, making sure the encoded image aligns with the standard clinical practice to avoid ambiguity. Third, the encoded prompts (either spatial info such as Point, Bbox, or Free-form text data), together with the encoded image, will be processed with mask decoder and output the expected masks.

Following Kaba et al. (2022); Mondal et al. (2023), we train a separate canonicalization network $h : \mathcal{X} \mapsto G$, where $\mathcal{X}$ represents the medical image sample space, $G$ represents the desired group, and $h$ is equivariant to $G$. This network generates group elements that transform input images into canonical frames, standardizing the image orientation before applying the prediction function. The Eq. 2 shows how this canonicalization process maps the transformed input back to a common space where the segmentation prediction network $p$ operates,

$$f(x) = \rho_{\text{out}}(h(x)) \, p(\rho_{\text{in}}(h^{-1}(x))x, \, \mathbf{t}) \tag{2}$$

Where $p$ is the segmentation prediction network (composed of the Image Encoder and Mask Decoder in Figure 12), $\mathbf{t}$ is the text prompt embedding produced by our text encoder, and $\rho_{\text{in}}$ and $\rho_{\text{out}}$ are input and output representations. The segmented images or masks produced by $p$ can be transformed back with $\rho_{\text{out}}(h(x))$ as needed. Without this transformation, $f$ is invariant; otherwise, it is equivariant. Thus, the FLanS architecture visualized in Figure 12 is invariant. We use ESCNN (Cesa et al., 2022) to build the canonicalization network. This approach has the advantage of removing the constraint from the main prediction network and placing it on the network that learns the canonicalization function. Appendix A provides a detailed introduction of symmetry and equivariant networks.

As the entire architecture achieves invariance or equivariance through canonicalization, the model produces the same segmentation or consistently transforms the segmentation according to the transformed input. In other words, the model always segment the same areas of interest regardless of the image's orientation with the same text prompt. For example, as long as the ground truth "right kidney" mask of a CT image has been shown to the model once, no matter how the orientation of the CT image and the location of the right kidney changes, the model will always segment the same area.

However, without proper training, $h(x)$ might map different images to inconsistent canonical frames, causing a distribution shift in the inputs to the prediction network and affecting performance. Thus, training the canonicalization network togther with the segmentation prediction network is essential to ensure consistent mapping to the desired frame. It is worth noting that users can choose to disable the canonicalizer when working with anatomy-agnostic prompts, as the segmented organ may differ if the original image is not in the canonical frame. The decision depends on whether the user wants to segment the original or the canonicalized image, as the model will segment whatever image is fed into the image encoder based on the provided text prompts.

### 3.4 TRAINING STRATEGY

We employ a three-stage training strategy for FLanS: 1) **Learning canonicalization**: we train the canonicalization network independently using FLARE22 training samples applied with random transformations from the O(2) group. The network is optimized using MSE loss between the canonicalized samples and their original counterparts. This encourages the canonicalization network to map

transformed samples back to their canonical orientations as seen in the FLARE22 dataset, preventing it from selecting arbitrary orientations that could degrade the performance of the prediction network. 2) **Learning text-prompted segmentation**: we train `FLanS` with the queries from Generator **G** as introduced in Section 3.1, without the canonicalization network on the original scans, using both anatomy-informed and anatomy-agnostic prompts. This ensures that the segmentation network learns to respond accurately to different types of prompts without interference from canonicalization and data augmentation. 3) **Learning augmentation and alignment**: In the final stage, we perform joint training on all scans, applied with random O(2) transformations. Since the canonicalization network may not always generate the exact canonical orientation the segmentation network is accustomed to in the beginning, this serves as a form of free augmentation for the segmentation networks. Over time, the canonicalization and segmentation networks align.

## 4 EXPERIMENT

### 4.1 DATASETS AND EXPERIMENTS SETUP

**Image Datasets**   To develop an effective organ segmentation model, we collected 1,437 CT scans from 7 public datasets, covering 24 partially labeled organs. Of these, 1,089 scans from MSD Antonelli et al. (2022), BTCV (Gibson et al., 2018), WORD (Luo et al., 2021), AbdomenCT-1K (Ma et al., 2022), FLARE22 (Ma et al., 2023), and CHAOS (Kavur et al., 2019) are used for training. The rest 65 scans, consisting of 10% of the FLARE22 dataset (in-domain), the official validation set of WORD (in-domain), and the official test set of RAOS (Luo et al., 2024) (out-of-domain), were used to evaluate model performance. To standardize the quality and reduce domain gaps across datasets, we applied pre-processing techniques such as slice filtering and intensity scaling to all CT scans. The finalized dataset comprised 91,344 images for training and validation, and 9,873 for testing. Detailed information on the dataset statistics and pre-processing steps are in Appendix B.

**Text Datasets**   Our text dataset was constructed using two types of queries: anatomy-agnostic and anatomy-informed. First, for each image, we identified organs corresponding to 6 representative positions: leftmost, rightmost, topmost, bottom, smallest, and largest. For each of these 6 position indicators, 100 anatomy-agnostic queries were generated, resulting in a set of 600 queries to serve as anatomy-agnostic segmentation prompts.[3] Second, for each organ, we generated 480 anatomy-informed queries in an expertise-driven style using the RAG query generator. By combining both anatomy-agnostic and anatomy-informed queries, we formed a text dataset comprising 12,120 unique queries for model training. During testing, a comprehensive text set was used, containing both in-domain and out-of-domain queries. Specifically, we generated 30 RAG-generated expertise-style queries (25%, in-domain), 30 human-generated non-expertise-style queries (25%, out-of-domain), and 60 RAG-generated non-expertise-style queries (50%, out-of-domain) for each organ, forming a test set of 120 queries per organ and 2,880 queries across all organs. Detailed information on the generation of the text queries is in Appendix B.

**Experiment Setup**   All experiments were conducted on an AWS ml.p3dn.24xlarge instance equipped with 8 V100 GPUs, each with 32 GB of memory. We used a batch size of 16 and applied the CosineAnnealingLR learning rate scheduler, initializing the learning rate for all modules at 0.0001. The AdamW optimizer was employed for training. A small $D_8$ equivariant canonicalization network was used, consisting of 3 layers, a hidden dimension of 8, and a kernel size of 9. To maintain consistency across the input and output formats, all scans from different datasets were resized to $1024 \times 1024$ and both predicted and ground truth masks were resized to $256 \times 256$ for fair comparison. For images with a single channel, the channel was duplicated to 3. All models' performance on the test sets is reported using both the Dice coefficient (Taha & Hanbury, 2015) and normalized surface distance (Heimann & Meinzer, 2009) .

### 4.2 ANATOMY-INFORMED SEGMENTATION

We first compare our model, `FLanS`, with the SOTA baselines on a held-out subset of the FLARE22 training set (FLARE), the public WORD validation set (WORD), and RAOs cancer CT images (RAOS). Both FLARE22 and WORD serve as in-domain test sets, while RAOS is an out-of-domain

---

[3]To ensure accurate position-to-organ mapping, position-driven organ-agnostic queries were applied only to images containing more than nine labeled organs during training.

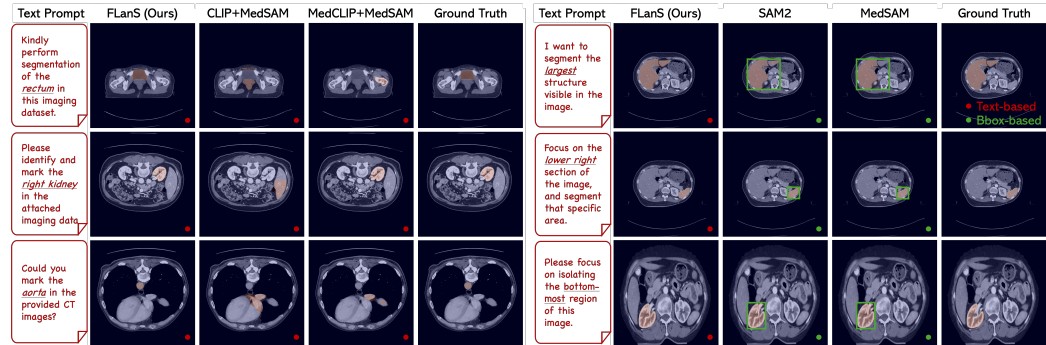

Figure 5: **Left**: Segmentation with anatomy-**informed** prompts. We could observe that `FLanS` can precisely segment the organ described in free-form text prompts, while other baselines make mistakes in identifying the organs. **Right**: Segmentation with anatomy-**agnostic** prompts. We could observe that the `FLanS` is texture-aware, descriptions of the sizes and positions can be understood, and is competitively accurate to the direct Bbox segment.

Table 2: **Anatomy-Informed** Segmentation Results: `FLanS` consistently outperforms baselines on both organ name and free-form text prompts segmentation tasks, demonstrating superior language understanding and segmentation accuracy across in-domain and out-of-domain datasets, even when applied with random transformations.

| Organ Name | FLARE | | WORD | | RAOS | | TransFLARE | | TransWORD | | TransRAOS | |
|---|---|---|---|---|---|---|---|---|---|---|---|---|
| | Dice | NSD | Dice | NSD | Dice | NSD | Dice | NSD | Dice | NSD | Dice | NSD |
| CLIP+MedSAM | 0.473 | 0.518 | 0.411 | 0.446 | 0.475 | 0.440 | 0.388 | 0.417 | 0.357 | 0.437 | 0.352 | 0.399 |
| MedCLIP+MedSAM | 0.557 | 0.516 | 0.466 | 0.510 | 0.419 | 0.320 | 0.485 | 0.415 | 0.342 | 0.378 | 0.336 | 0.336 |
| Universal Model | 0.649 | 0.697 | 0.512 | 0.408 | 0.442 | 0.301 | 0.380 | 0.290 | 0.299 | 0.278 | 0.200 | 0.201 |
| **FLanS** | **0.908** | **0.956** | **0.837** | **0.884** | **0.852** | **0.883** | **0.898** | **0.949** | **0.835** | **0.875** | **0.847** | **0.879** |

| Free Form | FLARE | | WORD | | RAOS | | TransFLARE | | TransWORD | | TransRAOS | |
|---|---|---|---|---|---|---|---|---|---|---|---|---|
| | Dice | NSD | Dice | NSD | Dice | NSD | Dice | NSD | Dice | NSD | Dice | NSD |
| CLIP+MedSAM | 0.425 | 0.468 | 0.381 | 0.347 | 0.402 | 0.400 | 0.342 | 0.434 | 0.356 | 0.456 | 0.339 | 0.357 |
| MedCLIP+MedSAM | 0.696 | 0.557 | 0.473 | 0.518 | 0.365 | 0.424 | 0.483 | 0.501 | 0.239 | 0.241 | 0.307 | 0.331 |
| Universal Model | — | — | — | — | — | — | — | — | — | — | — | — |
| **FLanS** | **0.912** | **0.958** | **0.830** | **0.889** | **0.854** | **0.885** | **0.896** | **0.942** | **0.833** | **0.888** | **0.865** | **0.899** |

test set, as neither our model nor the baselines were trained on this dataset. Although the original test sets already contain scans with varying orientations, we further evaluated the models' robustness by applying random O(2) transformations to the three test sets, creating additional sets: TransFLARE, TransWORD, and TransRAOS. More importantly, we tested the models using Anatomy-Informed text prompts, which included two types: purely organ names and free-form text descriptions.

As for the baselines, the `Universal Model` (Liu et al., 2023b) is the only published medical imaging foundation model that considers free-form text descriptions. This model integrates text description embeddings during training, while segmentation at the testing and inference stages is performed using organ IDs. Consequently, we evaluate this model with prompts consisting solely of organ names. Another widely used approach for text-prompt segmentation involves combining CLIP-based models (Radford et al., 2021) with segmentation models (Li et al., 2024; Wang et al., 2024). In these methods, segmentation models first generate potential masks based on a set of random bounding box or point prompts that span the entire image. CLIP-based models then embed both the text prompt and the cropped images from these masks. The final mask is selected based on the highest similarity between the cropped image embedding and the text embedding. To cover this approach, we include two additional baselines: 1) `CLIP + MedSAM`, where MedSAM (Wu et al., 2023) is SAM (Kirillov et al., 2023) fine-tuned on medical imaging datasets; and 2) `MedCLIP + MedSAM`, where MedCLIP (Wang et al., 2022d), a contrastive learning framework trained on diverse medical image-text datasets, is paired with MedSAM for segmentation.

As we can see from Table 2, `FLanS` achieves superior performance in segmenting based on organ name. More importantly, `FLanS` significantly outperforms the baselines on free-form text prompts segmentation, where the baselines struggle with more complex language input. This suggests that

training with diverse free-form text prompts enhances the model's ability to understand language and the relationship between text descriptions and medical images. Furthermore, `FLanS` maintains high Dice and NSD scores on the transformed test sets thanks to the help of the canonicalization network. The left panel of Fig. 5 visualizes the segmentations generated by the best baseline and `FLanS`, alongside their corresponding text prompts, illustrating our model's superior language understanding and segmentation accuracy.

## 4.3 ANATOMY-AGNOSTIC SEGMENTATION

To evaluate our model's ability to understand anatomy-agnostic text prompts, we tested its segmentation performance using prompts that contain only positional or size-related information. To the best of our knowledge, no existing model is designed to handle anatomy-agnostic text prompts. Therefore, we chose state-of-the-art `MedSAM` (Wu et al., 2023) (SAM fine-tuned on medical imaging datasets) and the latest `SAM2` (Ravi et al., 2024)

Table 3: **Anatomy-Agnostic** Segmentation Results: Comparison of `FLanS` using positional and size information text prompts vs. `MedSAM` and `SAM2` using Bboxes or points. `FLanS` achieves competitive or superior performance across both in-domain and out-of-domain test sets.

| Methods | FLARE | | WORD | | RAOS (*OOD*) | |
|---|---|---|---|---|---|---|
| | Dice | NSD | Dice | NSD | Dice | NSD |
| SAM2-large (*Point-prompt*) | 0.585 | 0.652 | 0.534 | 0.551 | 0.488 | 0.497 |
| SAM2-large (*Bbox-prompt*) | 0.873 | **0.906** | 0.848 | 0.802 | 0.818 | 0.749 |
| MedSAM (*Bbox-prompt*) | **0.887** | 0.872 | 0.783 | 0.781 | 0.697 | 0.681 |
| **FLanS** (*Free-form text*) | 0.844 | 0.841 | **0.855** | **0.853** | **0.851** | **0.850** |

as baselines. However, instead of text prompts, these models were provided with ground-truth organ Bboxes or point prompts. Our goal in this experiment is for `FLanS` to achieve comparable results to the baselines because `FLanS` is only given text prompts with positional or size information while the baselines are given the bounding box or point prompts of ground truth organ.

As shown in Table 3, `FLanS` the best or second-best performance across both in-domain and out-of-domain test sets. `MedSAM` performs well on the FLARE and WORD test sets but struggles on the RAOS test set due to the lack of training on that dataset. `SAM2`, when provided with bounding box prompts, consistently performs well across all test sets and demonstrates strong generalizability. However, its performance significantly degrades with point prompts, likely because medical scans lack the distinct edges present in the datasets `SAM2` was originally trained on. The right panel of Fig. 5 visualizes the segmentations produced by the best baseline and `FLanS`, along with their corresponding anatomy-agnostic text prompts. It demonstrates that `FLanS` can reliably segment the correct organs based on the provided positional or size information, such as *largest* and *lower right*.

## 4.4 ABLATION STUDY ON THE MODEL ARCHITECTURE

*Text Prompt: Highlight the right side renal organ.*

With Canonicalization      Without Canonicalization

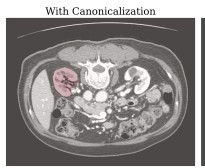 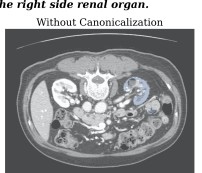

Figure 6: The model without canonicalization incorrectly highlights the left kidney due to confusion between anatomical position ("right kidney") and the organ's appearance on the right side of the image.

Table 4: Ablation study: prediction performance of `FLanS` and its variants with progressively removed components on the FLARE22 original and transformed test sets. Each row represents a version of the model with one additional component removed.

| Model Variants | Canonicalized Test Set | | Transformed Test Set | |
|---|---|---|---|---|
| | Dice | NSD | Dice | NSD |
| **FLanS** (full model) | **0.901**±**0.003** | **0.953**±**0.008** | **0.895**±**0.010** | **0.951**±**0.002** |
| – Canonicalization | 0.865±0.010 | 0.896±0.011 | 0.685±0.012 | 0.728±0.014 |
| – Data Augmentation | 0.883±0.012 | 0.930±0.017 | 0.289±0.011 | 0.328±0.019 |
| – Trainable ImgEncoder | 0.748±0.009 | 0.845±0.016 | 0.301±0.009 | 0.283±0.017 |
| – Classification Loss | 0.718±0.036 | 0.831±0.029 | 0.271±0.020 | 0.234±0.049 |

We conducted an ablation study of `FLanS` on the FLARE22 dataset (Ma et al., 2023) to understand the contribution of each component, as presented in Table 4. Using an 80%-10%-10% train-validation-test split on the public FLARE22 training set, we evaluate the models' performance on both the held-out test set and a transformed test set, which contained samples applied with random transformations from $O(2)$. Table 4 shows the prediction performance of `FLanS` and its variants, with components progressively removed. The results highlight that each component plays a crucial role in the model's overall performance. Notably, while data augmentation improved the model's robustness to random

transformations, it slightly reduced performance on the canonical test set, as the model had to handle various transformations. However, by canonicalization network, the segmentation backbone focuses specifically on canonicalized medical images, thus achieving the best performance on both test sets.

## 4.5 Effective Understanding of Free-Form Text Prompts

Fig. 7 left visualizes the t-SNE embeddings of free-form text prompts corresponding to all 13 FLARE22 data classes, including liver, right kidney, spleen, and others. The text prompt encoder effectively clusters these prompts, revealing anatomically structured semantics. This demonstrates `FLanS` has a strong capability in understanding and distinguishing free-form text prompts.

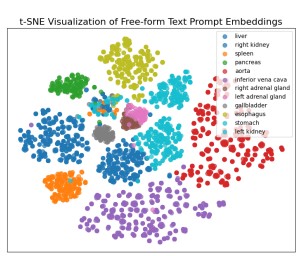 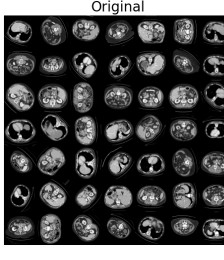 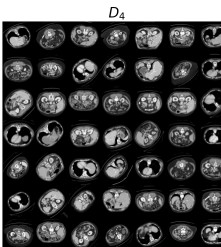 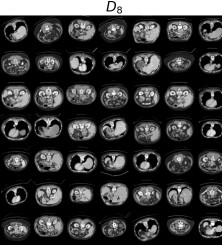

Figure 7: Left: t-SNE visualization of the free-form text prompt embedding space. Our method can effectively distiguish between different organ related queries. Right: Canonicalized CT scans from $D_4$ and $D_8$ canonicalization networks for a batch of randomly transformed scans from the FLARE22 dataset. Medical images can be successfully transformed back to an aligned canonicalization space.

## 4.6 Effectiveness of the Canonicalization

The right side of Fig. 7 shows the canonicalized CT scans from $D_4$ and $D_8$ canonicalization networks for a batch of original scans from the FLARE22 dataset applied with random transformations from $O(2)$ group. As the group order of the canonicalization network increases, the scans become more consistently aligned to a particular canonical orientation. The canonicalization networks use a shallow architecture with three layers, a hidden dimension of 8, and a kernel size of 9, demonstrating that even a simple network with a larger kernel can effectively achieve canonicalization.

More importantly, applying canonicalization before feeding the scans into the main segmentation network and making the entire architecture equivariant or invarianthelps prevent confusion caused by positional terms in the organ name. A text-prompt segmentation model understands positional cues such as "*left*" vs "*right*" but it may get confused between the anatomical position and the organ's appearance in the scan. For example, Fig. 6 shows segmentation predictions from models with and without canonicalization, given the anatomy-informed text prompt, "Highlight the right renal organ." Since the CT scan is not in the standard orientation, the right kidney appears on the left side of the image. Without canonicalization, the non-equivariant model incorrectly segments the left kidney, which appears on the right side. Our model can make consistent predictions of the right kidney regardless of the scan's orientation, allowing it to focus on learning the critical features of the organs.

## 5 Conclusion

In this work, we presented `FLanS`, a novel medical image segmentation model capable of handling diverse free-form text prompts, including both anatomy-informed and anatomy-agnostic descriptions. By integrating equivariance, our model ensures accurate and consistent segmentation across varying scan orientations, addressing a critical challenge in medical imaging. We also developed a RAG query generator for both realistic and synthetic prompt generation, and trained `FLanS` on over 100k medical images from 7 public datasets, covering 24 organ categories. `FLanS` outperforms baselines in both in-domain and out-of-domain tests, demonstrating superior language understanding and segmentation accuracy. Future works including extend `FLanS` to multi-organ segmentation tasks and further enhance RAG generator with multimodal data.

## Reproducibility Statement

Our code and dataset details are available at this anonymous repository [4]

---

[4] `https://anonymous.4open.science/r/SegmentAsYouWish-16F4/README.md`

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

# A  EQUIVARIANCE AND SYMMETRY

Equivariant neural networks are designed to explicitly incorporate symmetries that are present in the underlying data. Symmetries, often derived from first principles or domain knowledge, such as rotational or translational invariance, allow the network to process inputs in a way that is consistent with these transformations. This is particularly important when the ground truth functions respect such symmetries, as the incorporation of these properties can significantly enhance model performance and generalization.

**Group**  A group of symmetries or simply *group* is a set $G$ together with a binary operation $\circ\colon G \times G \to G$ called *composition* satisfying three properties: 1) *identity*: There is an element $1 \in G$ such that $1 \circ g = g \circ 1 = g$ for all $g \in G$; 2) *associativity*: $(g_1 \circ g_2) \circ g_3 = g_1 \circ (g_2 \circ g_3)$ for all $g_1, g_2, g_3 \in G$; 3) *inverses* if $g \in G$, then there is an element $g^{-1} \in G$ such that $g \circ g^{-1} = g^{-1} \circ g = 1$.

Examples of groups include the dihedral groups $D_4$ (symmetries of a square) and $D_8$ (symmetries of an octagon), as well as the orthogonal group $O(2)$, which represents all rotations and reflections in 2D space. Both $D_4$ and $D_8$ are discrete subgroups of $O(2)$.

**Representation**  A group representation defines how a group action transforms elements of a vector space by mapping group elements to linear transformations on that space. More specifically, a group representation of a group $G$ on a vector space $V$ is is a homomorphism: $\rho\colon G \to \mathrm{GL}(X)$, where $\mathrm{GL}(X)$ is the group of invertible linear transformations on $V$. This means for any $g_1, g_2 \in G$, $\rho$ is a linear transformation (often represented by a matrix) such that the group operation in $G$ is preserved:

$$\rho(g_1 g_2) = \rho(g_1)\rho(g_2) \tag{3}$$

**Equivariance**  Formally, a neural network is said to be equivariant to a group of transformations $G$ if applying a transformation from the group to the input results in a corresponding transformation to the output. Mathematically, for a function $f\colon X \to Y$ to be $G$**-equivariant**, the following condition must hold:

$$f(\rho_{\mathrm{in}}(g)(x)) = \rho_{\mathrm{out}}(g)f(x) \tag{4}$$

for all $x \in X$ and $g \in G$, where $\rho_{\mathrm{in}}\colon G \to \mathrm{GL}(X)$ and $\rho_{\mathrm{out}}\colon G \to \mathrm{GL}(Y)$ are input and output representations (Bronstein et al., 2021). Invariance is a special case of equivariance where the output does not change under the group action. This occurs when the output representation $\rho_{\mathrm{out}}(g)$ is trivial. Figure 8 visualize how the equivariant and invariant networks work.

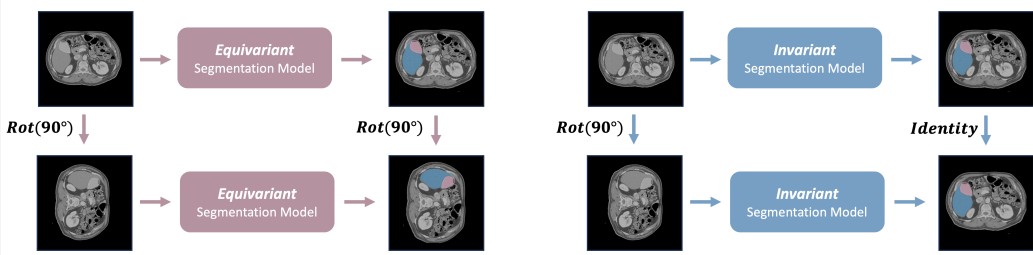

Figure 8: An equivariant model (left) ensures that its output transforms in a specific, predictable way under a group of transformations applied to the input, preserving the structure of the transformation (e.g., rotating the input results in a correspondingly rotated output). In contrast, an invariant model (right) produces an output that remains unchanged regardless of any transformations applied to the input from the same group.

**Equivariance via weight-sharing**  One of the primary approaches to incorporating symmetry into neural networks is through weight sharing (Satorras et al., 2021; Cohen et al., 2018; Wang et al.). This approach enforces equivariance by constraining the network's architecture so that the weights are shared across different group elements. For example, in $G$-convolutions (Cohen & Welling, 2016), the same set of weights is shared across the transformed versions of the input, ensuring that the network's predictions remain consistent under those transformations. In a layer of $G$-steerable

CNNs (Weiler & Cesa, 2019), a set of equivariant kernel bases is precomputed based on the input and output representations, and the convolution kernel used is a linear combination of this equivariant kernel basis set, where the coefficients are trainable. Similar approaches can also be used to develop equivariant graph neural networks (Geiger & Smidt, 2022). These architectures directly modify the network's layers to be equivariant, ensuring that each layer processes symmetries in a way that is aligned with the desired group. While powerful, this approach imposes architectural constraints, which may limit the flexibility of the network and prevent leveraging large pretrained models.

**Equivariance via canonicalization**   An alternative to weight sharing is incorporating symmetry through canonicalization (Kaba et al., 2022; Mondal et al., 2023), where, instead of modifying the network's architecture to handle symmetries, the input data is transformed into a canonical form. In this approach, a separate canonicalization network, which is itself equivariant, preprocesses the input, transforming it into a standard, or canonical, representation. This canonicalized input is then passed to a standard prediction network that does not need to be aware of the symmetries. If the corresponding inverse transformation is applied to the output of the prediction network, the entire model becomes equivariant; otherwise, the model remains invariant. This method has several advantages. First, it does not require altering the architecture of the prediction network, allowing for the use of large pre-trained models without modification. Second, by ensuring that the input data is in a canonical form, the prediction network only needs to learn the mapping from the canonical input to the output, without needing to learn all transformed samples. This can lead to improved performance and robustness, especially in scenarios where the prediction task does not naturally align with the symmetry group or where architectural constraints might hinder performance. Thus, in our work, we leverage canonicalization to achieve equivariance in the segmentation task. By transforming the input into a canonical form using a simple equivariant canonicalization network, we ensure that our prediction network remains unconstrained and can fully utilize its capacity for learning without the need for architectural modifications. This approach offers the benefits of symmetry-aware processing while maintaining the flexibility and power of unconstrained neural network architectures.

# B   DETAILED DATASET DESCRIPTION

**Image Data Collection and Preprocessing**   For model development and evaluation, we collected 1,437 CT scans from 7 public datasets. A detailed summary of the datasets is provided in Table 5. In total, 24 organs are labled in the assembled datasets, with a strong focus on segmentation targets in the abdominal region. The organ class distribution across the datasets is shown in Fig 9. To standardize quality and reduce domain gaps, we applied a preprocessing pipeline to all datasets. Specifically, we mapped the Hounsfield unit range [-180, 240] to [0, 1], clipping values outside this range. To address dimension mismatches between datasets, masks, and images, all scans and masks were resized to $1024 \times 1024$. The 3D scan volumes were sliced along the axial plane to generate 2D images and corresponding masks. To ensure labeling quality, organ segments with fewer than 1,000 pixels in 3D volumes or fewer than 100 pixels in 2D slices were excluded. The finalized dataset consisted of 101,217 images, with 91,344 (90.25%) used for training and validation, and 9,873 (9.75%) reserved for testing.

Table 5: Overview of the datasets used in this study.

| Dataset | # Training scans | # Testing scans | Annotated organs[1] |
|---|---|---|---|
| AbdomenCT-1K | 722 | — | Liv, Kid, Spl, Pan |
| MSD[2] | 157 | — | Lun, Spl |
| WORD | 100 | 20 | Liv, Spl, LKid, RKid, Sto, Gal, Eso, Pan, Duo, Col, Int, LAG, RAG, Rec, Bla, LFH, RFH |
| FLARE22 | 40 | 5 | Liv, RKid, Spl, Pan, Aor, IVC, RAG, LAG, Gal, Eso, Sto, Duo, LKid |
| CHAOS | 40 | — | Liv |
| BTCV | 30 | — | Spl, RKid, LKid, Gal, Eso, Liv, Sto, Aor, IVC, PVSV, Pan, RAG, LAG |
| RAOS[3] | — | 40 | Liv, Spl, LKid, RKid, Sto, Gal, Eso, Pan, Duo, Col, Int, LAG, RAG, Rec, Bla, LFH, RFH, Pro, SV |

**Test Data Creation**    Different from existing work that solely chases for a higher segmentation accuracy, in this paper, we expect to evaluate the segment model's performance in dual tasks: The free-form text understanding ability and segmentation ability.

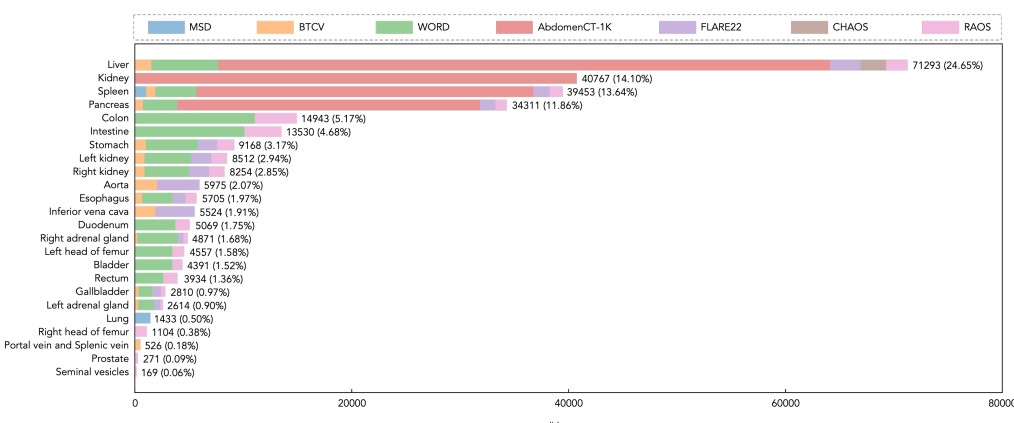

Figure 9: Distribution of labeled organs across the collected datasets. The image count for each organ and its corresponding ratio is marked in the plot.

In order to verify the model's ability to understand the language descriptions, we construct a query dataset (test set) from two resources: 1. Real-world human queries; 2. LLM-generated synthetic queries. For the first kind of real-world queries, we have two groups of annotators, **Domain Expert** and **Non-Expert**. Domain experts are from clinical hospitals who provide the query materials from their daily diagnosis notes, this group of people tends to use professional vocabulary, and their intention might not be explicitly expressed in a professional report, such as in the report, the doctor writes 'Concerns in the hepatic area that warrant a more focused examination', which implicitly means the 'liver is the area of interest under certain symptom'. Another group of query providers is the non-expert, who are not specialized in clinical or equipped with medical specialties. We explain to this group of people that their task is to write a sentence and show the intention of segmenting the target organ/tissue in a CT scan, e.g., the liver. This aspect of real queries represents a more general and non-specialist approach to expressing the need for segmentation (such as in the student learning scenarios). Apart from real query data, we incorporate synthetic test queries to enlarge the test samples and add randomness in various expressions. The synthetic test is generated by GPT-4o following the template shown below:

---

**The Prompt Template to Generate Synthetic Queries.**

- - - - - - - - - - - - - - - - - - - - - - - - - - - - - - - - - - - - - - - - - - - - - - - - - - - - - - -

**System Description:** You are a doctor with expert knowledge of organs.

**Task Description:** Now you are making a diagnosis of a patient on the CT scan over {body part}. You find a potential problem on {organ name} and want to see more details in this area, please query for segmentation by free-form text. Please make sure to deliver the segment target explicitly, and you are encouraged to propose various expressions.

**Format:** {segmentation query}, {explain reason}.

**Example:** Given that, {body part} is `abdomen` and {organ name} is `liver`.

---

[1]For simplicity, the following abbreviations are used: Liv (liver), Kid (kidney), Spl (spleen), Pan (pancreas), Col (colon), Int (intestine), Sto (stomach), LKid (left kidney), RKid (right kidney), Aor (aorta), Eso (esophagus), IVC (inferior vena cava), Duo (duodenum), RAG (right adrenal gland), LHF (left head of femur), Bla (bladder), Rec (rectum), Gal (gallbladder), LAG (left adrenal gland), RHF (right head of femur), PVSV (portal vein and splenic vein), Pro (prostate), and SV (seminal vesicles).

[2]Only the lung and spleen subsets from MSD were used.

[3]We used CancerImages (Set1) from RAOS as our out-of-domain test set. To avoid overlap, any scans in RAOS that were extended from WORD were excluded from testing.

Your response should be something like: {Please identify the liver for me for more analysis.} {Because elevated liver enzymes alanine aminotransferase (ALT) in the blood tests might indicate liver inflammation or damage}.

**Output:** {Placeholder}

---

The overall structure of the test dataset is shown in Figure 10. It consists of 25% expert queries, 25% normal queries, and half synthetic queries. In total, we have 2880 (24 organs x10 queries x3 x2x2) text queries. Each of the queries is labeled with the correct organ name to segment. This will be used to evaluate the ability of our learned TextEncoder model to understand correct intentions based on free-form language description.

At the same time, the organ names are connected to another segmentation test set, which contains several (how many) medical images such as CT scans, MRIs, etc. And stand on the results of interest-category identification, we conduct further segmentation result analysis, including the normal segmentation precision study, and also the equivariant identified segmentation study.

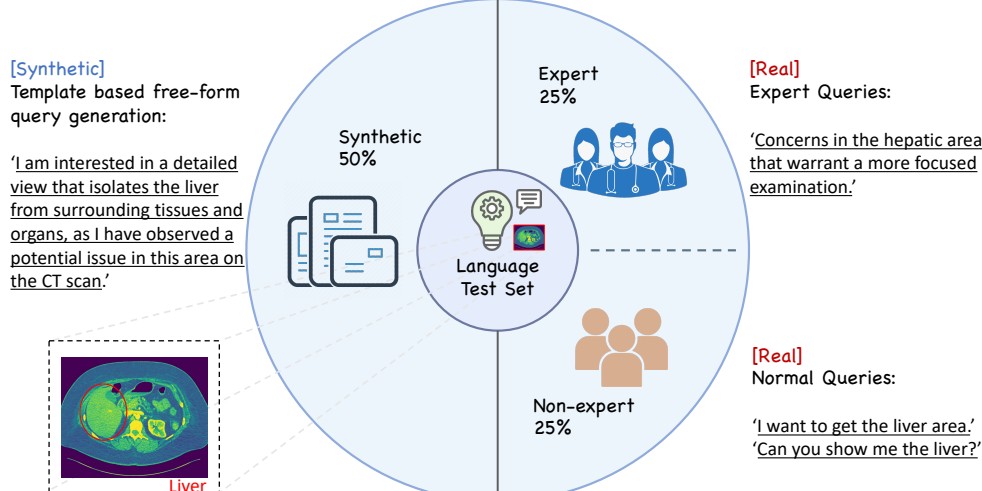

Figure 10: The Language Test Set for Verifying the Query Understanding Ability. It contains three aspects of components, real data - expert group, real data - non-expert group, and synthetic data.

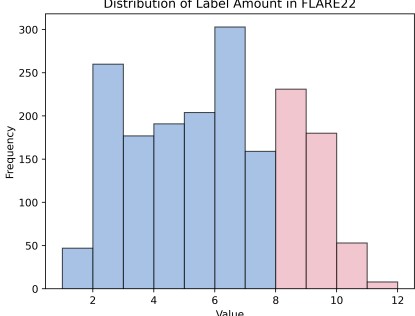

Figure 11: Positional prompt dataset provider split, we take the slices with more than $\alpha$ labels, where we set $\alpha = 8$ in this illustration (while 13 is the total label amount) as a split threshold, ensure that the slice used for training the label-agnostic provides sufficient semantics in the image content, such as left, upmost or largest, etc. Similarly, we process the other datasets such as BTCV and WORD.

## C  TRAINING DETAILS

In the training process, we provide details of the configuration files and instructions below:

All experiments were conducted on an AWS ml.p3dn.24xlarge instance equipped with 8 V100 GPUs, each with 32 GB of memory. We used a batch size of 16 and applied the CosineAnnealingLR learning rate scheduler, initializing the learning rate for all modules at 0.0001. The AdamW optimizer was employed for training. A small $D_8$ equivariant canonicalization network was used, consisting of 3 layers, a hidden dimension of 8, and a kernel size of 9. To maintain consistency across the input and output formats, all scans from different datasets were resized to $1024 \times 1024$ and both predicted and ground truth masks were resized to $256 \times 256$ for fair comparison. For images with a single channel, the channel was duplicated to 3. All models' performance on the test sets is reported using both the Dice coefficient (Taha & Hanbury, 2015) and normalized surface distance following the existing work (Heimann & Meinzer, 2009).

## D  ATTENTION MAP

In this section, we demonstrate the attention map to visualize the correlation between text embedding and image embedding.

These attention maps based on atonamy-agnostic prompts not only address the first reviewer's query about plotting the relationship between text and images but also counter the second reviewer's concern that 'our method seems capable of inferring which organ is desired given a text prompt and images.

For example, consider the attention map generated with the prompt 'segment the part located at the topmost portion' it does not highlight just one organ. Instead, all organs at the top are highlighted. This demonstrates that our model is not merely overfitting data to infer a specific organ; rather, it has a deep understanding of the text and its relationship to the medical image.

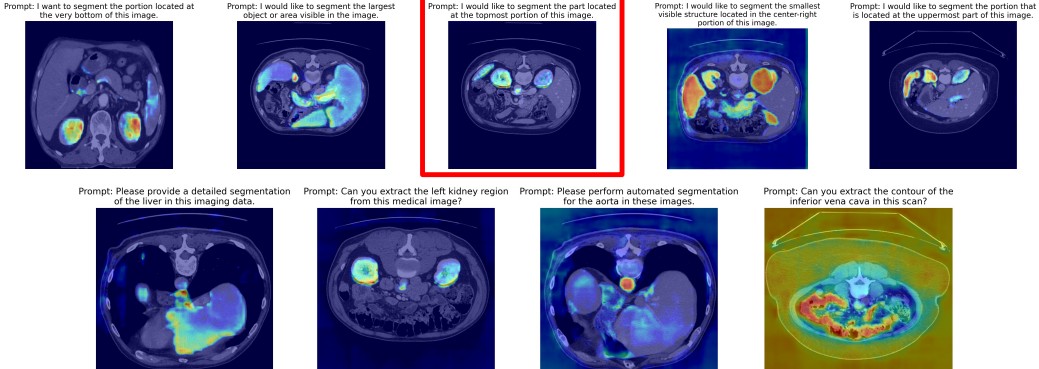

Figure 12: The examples of attention maps in FLanS segmentation tasks. The attention maps are computed based on the scaled product of text embedding from the text encoder and image embedding from the image encoder.

## E  MORE USE CASE DEMONSTRATIONS

In this sections, we provide more examples of segmentation results, ranging from the Anatomy-Informed to Anatomy-Agnostic ones. In Anatomy-Informed segmentation, we conduct two versions of illustration, first, we show the simplest (organ name is explicitly described in the prompt) segmentation, as shown in Fig. 15, and to add on more complexity and showcase how FLanS is beneficial for real-world clinical use cases, we take the diagnosis data from pseudonymized real EMR (eletronic medical record) data in Fig. 13, FLanS is able to detect the organ from the long and redundant descriptive symptoms and provide accurate segmentation, this is especially useful for providing diagnosis assistance based on doctors' notes. And last, we show size lated Anatomy-Agnostic segmentations as in Fig. 14.

Figure 13: The implicit Anatomy-Informed demonstration on a deployed version of the `FLanS` model, it provides real-time inference ability and can be robust to any format type of the prompts, either lengthy or redundant, it can still perform effectively to identify the most cared organ. In this image, two examples of a real EMR record data is provided (personal information such as name and age is pseudonymized.) We could observe that, even sometimes the actual label is not explicitly described as in upper half of this image, the model can perform segmentation accurately because it has seen the semantic similar corpus in the training period, and aligned these symptom related texts with the correct segmentation area.

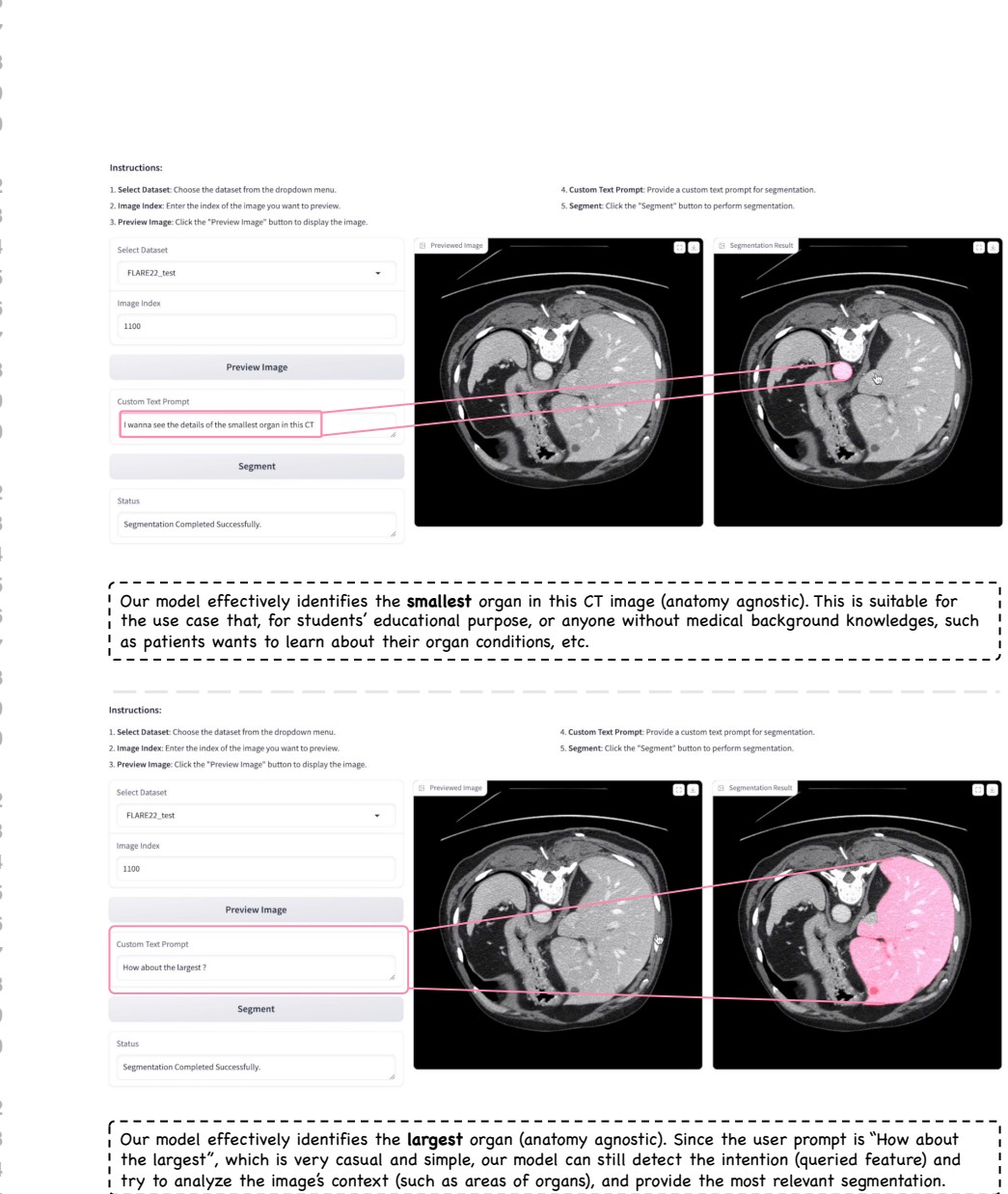

Figure 14: The Anatomy-Agnostic demonstration on a deployed version of the `FLanS` model, this feature is specially designed for a larger group of users who cares about the medical image scannings, but lacks the professional background knowledge, such as students, and patients. This supports the segmentation with the size relevant or positional relevant prompts, the above figures shows that `FLanS` is able to successfully segment those largest, smallest organs in the provided scans.

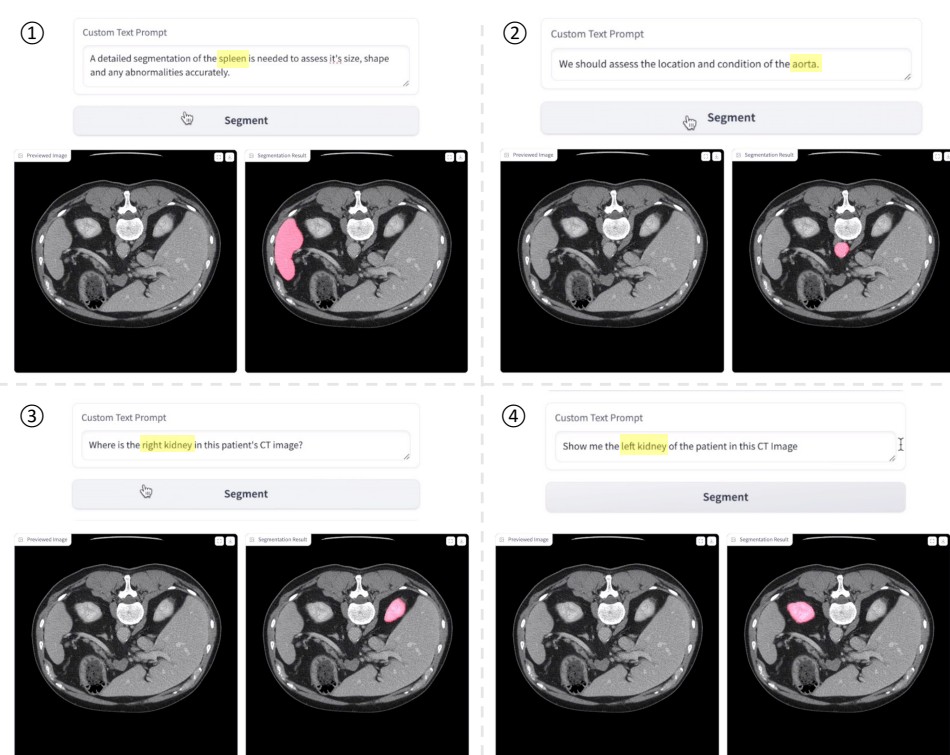

Our model effectively identifies the **anatomy informed (with explicit organ names)** in this image. This was a simple task when solely performed segmentation using pure label text, however, the traditional methods is easily influenced by the redundant information or extra descriptions, leading to a confused intention and cause segmentation performance drop. In comparison, our method can flexibly understand the segment-intention with any kind of text descriptions, and performs comparatively to those trained on simple labels, providing more convenient prompt interactions.

Figure 15: The explicit Anatomy-Informed demonstration on a deployed version of the FLanS model, in this figure, we showcase four examples of segmentation results by organ names relevant prompts. In the prompt content, it mentions the organ names, so as to instruct the segment action. As in the ① and ②, the model segment spleen and aorta successfully. And as shown in the ③ and ④, the model is also able to distinguish the right kidney and left kidney regardless of the angle and position of the scan is taken, enforced by the canonicalization module as introduced in the Section 3.3.

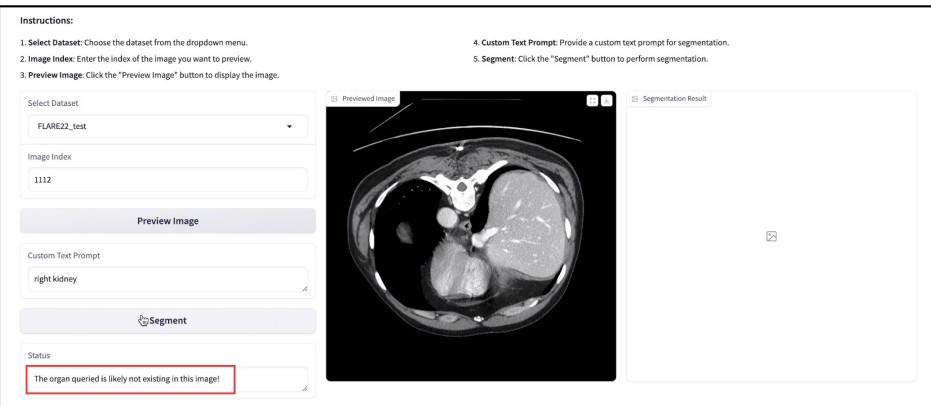

Figure 16: This image provides an extra analysis on the corner case detection. As shown in this image, there exist situations in blurry, low-quality meta data leading to the invisibility of an organ, or that an organ does not exist. FLanS is able to detect such cases and provide feedback that *The organ queried is likely not existing in this image!*. This is realized by a filter layer of function upon the predicted Probability for an organ area, the threshold is set to $\alpha = 0.5$. One can easily tune the parameter based on the actual require confidence of the FLanS.

