# OpenReview forum: "Segment as You Wish: Free-Form Language-Based Segmentation for Medical Images"
_ICLR.cc/2025/Conference — ICLR 2025 Conference Withdrawn Submission_

### Official Review · Reviewer_zBA8 · 2024-10-27

**Soundness:** 3
**Presentation:** 3
**Contribution:** 2
**Rating:** 5
**Confidence:** 5

**Summary:**

The paper presents FLanS, a novel medical image segmentation model that can understand and respond to free-form text prompts. It features a RAG-based text prompt generator and a symmetry-aware canonicalization module.

**Strengths:**

FLanS is trained on over 100k images and demonstrates strong language understanding and segmentation accuracy across various datasets. The key contributions are:
A RAG-based generator for diverse text prompts.
The FLanS model for text-driven medical image segmentation.
A symmetry-aware module for consistent segmentation across different scan orientations.

**Weaknesses:**

1. Comparison to the state-of-the-art baselines from natural images in the tasks like RES(Referring Expression Segmentation)[1]. The paper claims superior performance over SOTA baselines. However, it would be beneficial to see more direct comparisons, including visual examples and error analysis, to better understand where and how FLanS outperforms existing methods.
2. Lack of theoretical analysis, as a paper submitted to ICLR.
3.More details of making such a dataset.


[1] Liu, Chang, Henghui Ding, and Xudong Jiang. "GRES: Generalized referring expression segmentation."Proceedings of the IEEE/CVF Conference on Computer Vision and Pattern Recognition. 2023.

**Questions:**

1. Comparison to the state-of-the-art baselines from natural images in the tasks like RES(Referring Expression Segmentation)[1]. The paper claims superior performance over SOTA baselines. However, it would be beneficial to see more direct comparisons, including visual examples and error analysis, to better understand where and how FLanS outperforms existing methods.
2. Lack of theoretical analysis, as a paper submitted to ICLR.
3.More details of making such a dataset.


[1] Liu, Chang, Henghui Ding, and Xudong Jiang. "GRES: Generalized referring expression segmentation."Proceedings of the IEEE/CVF Conference on Computer Vision and Pattern Recognition. 2023.

---

> ### Author Response · Authors · 2024-11-22
> **Replies to the reviwer's concerns**
>
> > 1. Comparison and discussion to other SOTA baselines.
>
> Thank you, we have updated the paper with a discussion on the SOTA method [1] in related work.
>
> **As for more direct comparisons**
>
> Since MedSAM has been taken as a prevalent method in the medical image segmentation domain, so we mainly compared ours with this work, as well as the most recently released SAM2, which we believe is the current very dominant method.
>
> Besides, as we surveyed by the time of our draft-paper, there are only few of the methods take the free-form language prompt for medical image segmentation tasks, we found most of the methods take fixed-label for segmentations and we tried to conduct two sets of experiments regarding the text-guided segmentation performance.
>
> -  Compare our model's performance by using the fixed labels for anatomy-informed tasks. This makes a fair comparison to those models that have limited capability in free-form language understandings, such as the `Universal Model` in Table 2. We fed the same labels that can be accepted by existing methods and proved that our method can achieve similar or even better performance, while FLanS maintains the ability to process arbitrary descriptions.
>
> -  Compare our model's performance to those methods that technically can be generalized to free-form text prompts. We realize this by adding a text-encoder to the original architecture (i.e., MedSAM), such as `CLIP+MedSAM` and `MedCLIP+MedSAM`, we tend to generalize it to free-form tasks, please refer to Table 2. We also showcase that our method consistently performs better.
>
> We will definitely include more comparisons to other baseline methods in the updated version of the paper with new methods emerging.
>
> Regarding the GRES and FlanS (ours), they address distinct challenges in their respective domains, making direct comparisons difficult. While GRES focuses on generalized referring expression segmentation for natural images, handling complex multi-target and no-target scenarios with innovative benchmarks and models, FlanS (ours) targets medical image segmentation, emphasizing clinical precision and usability. FlanS excels in integrating anatomical knowledge and supporting free-form language prompts, tailored specifically for healthcare applications, while GRES lacks the medical specificity and precision required for clinical tasks.
>
> >2. Lack of theoretical analysis.
>
> Thank you for the suggestion. However, the focus of this paper is not on theoretical contributions. Instead, it aims to introduce a novel RAG-enhanced training framework for text generation that improves text alignment with large foundation models. Additionally, we propose a conceptual distinction between two types of free-form language prompts: anatomy-agnostic and anatomy-informed that better tackle the properties of the two angles.
>
> >3. More details of making such a dataset.
>
> In the original paper, we presented detailed explanations regarding the construction of the dataset being used in this experiment, including the data resources, data type, and text-prompt generation template, please check the details in the `Appendix section B DETAILED DATASET DESCRIPTION`.
>
> **We would like to thank the reviewer again and will keep updating the draft accordingly to improve the paper's quality of expression and authority**.

---

> > ### Comment · Reviewer_zBA8 · 2024-11-24
> >
> > Dear Authors,
> >
> >
> > Thank you for your response to my previous comments. I have read your explanations with care, but I must say that I still find the algorithmic novelty lacking. I maintain my stance that the paper would be better suited for MICCAI, where the clinical application aspects could be more prominently featured and appreciated.

---

### Official Review · Reviewer_yQbg · 2024-10-29

**Soundness:** 2
**Presentation:** 3
**Contribution:** 1
**Rating:** 1
**Confidence:** 5

**Summary:**

The authors introduce FLanS, a medical image segmentation model capable of handling diverse free-form text prompts, including both anatomy-informed and anatomy-agnostic descriptions. By integrating equivariance, it ensures accurate and consistent segmentation across varying scan orientations. Trained on over 100,000 medical images from seven public datasets covering 24 organ categories, FLanS outperforms baselines in both in-domain and out-of-domain tests.

**Strengths:**

1.	The paper is well-written, and the core idea is easy to follow.
2.	The proposed method is evaluated on diverse organ segmentation datasets.

**Weaknesses:**

## 1. Insufficient Motivation for FLanS:
The authors propose FLanS, a model designed to perform organ segmentation in CT scans based on free-form text prompts. However, the motivation behind this approach appears weak for the following reasons:

(1) Existing Tools: The authors claim that in real-world scenarios, clinicians often rely on natural language commands (e.g., "Highlight the right kidney" or "Segment the largest organ"). However, tools like TotalSegmentator[1,2,3] already segment most organs in CT and MRI scans effectively. These tools are widely integrated into platforms such as 3D Slicer[4], allowing clinicians to obtain segmentation results without needing text-based interactions.

(2) Questioning the Need: Given that TotalSegmentator covers a vast range of anatomical structures, the need for a text-based interaction model seems redundant. The authors do not provide a clear explanation of what organs or structures remain uncovered by existing solutions.

## 2. Addressable Challenges Without FLanS
The authors identify another challenge related to orientation differences between preprocessed datasets:

Orientation Standardization: The authors argue that varying scan orientations pose a problem that requires a "Canonicalized Network." However, this issue can be readily addressed with widely available Python libraries such as SimpleITK and nibabel, which can standardize 3D medical image orientations. Training an additional network specifically for orientation seems unnecessary and overcomplicated.

**Questions:**

In conclusion, the paper’s primary motivation for proposing FLanS is questionable. The authors' claim of a real-world need for text-prompt-based segmentation lacks sufficient justification, given that existing tools already handle the tasks effectively. Additionally, the technical challenges outlined, such as orientation standardization, can be addressed through simpler, more established methods, making the proposed solution less compelling.

---

### Official Review · Reviewer_7cgt · 2024-10-29

**Soundness:** 2
**Presentation:** 2
**Contribution:** 2
**Rating:** 3
**Confidence:** 4

**Summary:**

### Summary:
This paper introduces a free-form language-based segmentation algorithm termed FLanS. This model allows language-prompt-based segmentation of 24 different organs by creating text prompts in a RAG fashion. They differentiate into two type of prompts, anatomy-informed and anatomy-agnostic text prompts that are supposedly commonly used in practice and should allow for easier segmentation given a new clinical image. The authors evaluate their method on a variety of abdominal organ CT downstream datasets, namely FLARE22, WORD and RAOS showing improved performance over the chosen baselines.

**Strengths:**

I agree with the authors that language inclusions are currently underutilized in the domain of medical image segmentation and I like their creation of additional text reports through the RAG approach. Moreover, the description of position to steer predictions is novel.

**Weaknesses:**

I am highly skeptical that the current capabilities of the method are introducing new capabilities over existing methods in any meaningful way, which I will elaborate on in the following.

### Main concerns:
While I believe the integration of language to steer segmentation is very useful, I believe the author’s current use-case is not convincing at all. The following are the main points of critique:
1. Every clinician or experienced user is able to easily state which organ they are interested in. There is no added benefit of trying to predict which organ one refers to from a report description, this raises questions about the anatomy-informed prompt segmentation and also compromises the contribution of the RAG component that emulates clinical prompts.
2. The anatomy-agnostic setting is also not useable in the current format: In every clinical setting patients are present in 3D format. Given that this method is 2D, the “largest” organ will not be consistent across all 2D slices when one does whole-volume inference. So if one wants to use this method the clinician/user would have to adapt the slices to infer or create unique prompts for unique slices, which is both very unpractical.
3. Due to the proposed method being closed-set and being constrained to basic organ segmentation, any supervised model that does organ segmentation (and predicts all organs) would currently have a very similar capability, without the prior issues. The user would just have to discard all segmentations he is not interested in.
4. Regarding canonicalization, I would like to see the prevalence of these cases. Generally, all 3D images come with meta-information in the image header that should allow to re-orient it in a canonical way already. Hence I would like to have a quantification of how often this overall occurs to convince me that this is actually a problem worth solving.
5. Evaluation: It currently seems like the authors are only testing their capabilities in a closed-set setting. The final anatomy-informed prompts are the same as they used during training. To actually show that their method provides novel capabilities experiments where they try to predict novel classes given the text guiding could be conducted, which would greatly improve the utility of their proposed method.
6. Baselines: Since this is very close to supervised organ segmentation I would like to see supervised performance as reference (At least as a baseline to know where the performance is relative to a supervised 3D Model.

### Recommended improvements:
In my opinion, the utility of the method hinges entirely on being able to apply it in an open-set setting (a setting where they predict novel structures that they did not train on.  The way to show this would be to:
A) Predict a withheld/additional abdominal organ that was not trained on or
B) Predict a novel anatomy-agnostic prompt that they did not train on. An example for B could be: “Predict the organ above the liver.” Something that was not seen before and that could show that the model actually has learned more than the artificially created prompts.
C) Change predictive behavior for a certain organ. E.g.: “Predict the liver, but without the hepatic vessels.”
Any of these use-cases would actually bring novel capabilities, that would not be achievable or solvable by a simple supervised model.
This would likely be much easier if the authors used pathologies and described the pathology "In the lower left lobe of the liver is a hyperintensity, please segment it for me" -- This would likely make it much easier to become open-set and do something that basic supervised methods cannot do.

Summarizing, I find the currently displayed capabilities and results not convincing, the author’s method seems to be capable of inferring which organ is desired given a prompt, which lacks overall motivation.

**Questions:**

Q1: What are the use cases where FLanS is preferable over a TotalSegmentator? / Why shouldn't I ask ChatGPT to predict the class label and then just have a script retrieve the TotalSegmentator mask instead? (For anatomy-informed prompts)
Q2: How did you account for the "Segment the leftmost organ" not changing when iterating along 2D slices? -- How would you imagine this prompt should be applied in practice?
Q3: Did you evaluate fully in 3D or did you only consider a 2D slice by itself?

If the authors are able to provide experiments regarding A, B or C in the rebuttal (should be do-able since it should be just about providing new prompts) or convince me that their free-form text organ segmentation shows benefits over a TotalSegmentator model I am certainly willing to raise my score.

---

> ### Author Response · Authors · 2024-11-24
> **Replies to the reviwer's concerns**
>
> We really appreciate it for the reviewer's comments! We will reply to the concerns and questions as follows.
>
> ### Concerns
>
> > 1. Every clinician or experienced user is able to easily state which organ they are interested in. There is no added benefit of trying to predict which organ one refers to from a report description, this raises questions about the anatomy-informed prompt segmentation and also compromises the contribution of the RAG component that emulates clinical prompts.
>
> We propose to handle `free form text prompt` in two aspects: **anatomy-informed** and **anatomy-agnostic**, as for this question, **anatomy-informed** is indeed to handle those scenarios that users can explicitly describe what organ they expect to segment, which is the simplest case, we generalize the use case to the diagnosis report that the complex clinical descriptions may not be reflecting specific organ name directly, such as "hydronephrosis" may refer to the "kidney" in the report. Our method can do well (as shown in the upper part of Table 2) in the simplest case this concern is about, and the RAG-based prompt generation is to enhance the ability in vague queries.
>
>
> > 2. The anatomy-agnostic setting is also not useable in the current format: In every clinical setting patients are present in 3D format. Given that this method is 2D, the “largest” organ will not be consistent across all 2D slices when one does whole-volume inference. So if one wants to use this method the clinician/user would have to adapt the slices to infer or create unique prompts for unique slices, which is both very unpractical.
>
> Our current 2D approach is an initial step towards more complex, fully integrated 3D segmentation capabilities. And the autonomy-agnostic prompts are currently specific to 2D slices, this is for the use case that non-professional practitioners want to conduct studies or understand the medical image based organ segmentations. So they could describe slice by slice descriptions on what is observed in certain presented CT scans.
>
> > 3. Due to the proposed method being closed-set and being constrained to basic organ segmentation, any supervised model that does organ segmentation (and predicts all organs) would currently have a very similar capability, without the prior issues. The user would just have to discard all segmentations he is not interested in.
>
> Even though the organ segmentation task itself has been an interest with multiple methods, this paper focuses on a free-form text trigger,  which offers a more tailored, interactive, and context-aware solution.
>
> > 4. Regarding canonicalization, I would like to see the prevalence of these cases. Generally, all 3D images come with meta-information in the image header that should allow to re-orient it in a canonical way already. Hence I would like to have a quantification of how often this overall occurs to convince me that this is actually a problem worth solving.
>
> Public datasets often lack metadata due to privacy concerns, making it challenging to adjust image orientation using standard methods. Our model's canonicalization approach automatically standardizes image orientation, eliminating the need for manual intervention and enhancing operational efficiency and scalability.
>
> > 5. Evaluation: It currently seems like the authors are only testing their capabilities in a closed-set setting. The final anatomy-informed prompts are the same as they used during training. To actually show that their method provides novel capabilities experiments where they try to predict novel classes given the text guiding could be conducted, which would greatly improve the utility of their proposed method.
>
> Thanks for the suggestions, but the training text prompts include the RAG-based generations based on real clinical reports, while the test includes 25% expert (clinician) generated, 25% non-expert human-generated, and 50% Synthetic (RAG/GPT generated) data as shown in Fig.10.
>
> We are willing to improve the test set and add more diverse prompts to verify the effectiveness of the natural language understanding ability of the proposed method.
>
> > 6. Baselines: Since this is very close to supervised organ segmentation I would like to see supervised performance as reference (At least as a baseline to know where the performance is relative to a supervised 3D Model.
>
> We have compared the `Universal Model` [1] as a baseline, in fact, most of the supervised organ segmentation method strictly requires the same format of the input label for segmentation, while these labels must be seen in the list of the training set, so we conducted the label-based segmentation (which is an easy task for our model), as shown in Table 2 (upper), our performance is better across all of the test datasets across the baselines.
>
> [1] Liu J, Zhang Y, Chen J N, et al. Clip-driven universal model for organ segmentation and tumor detection[C]//Proceedings of the IEEE/CVF International Conference on Computer Vision. 2023: 21152-21164.

---

> > ### Author Response · Authors · 2024-11-24
> > **Cont. of the replies**
> >
> > ### Questions
> >
> > > Q1: What are the use cases where FLanS is preferable over a TotalSegmentator? / Why shouldn't I ask ChatGPT to predict the class label and then just have a script retrieve the TotalSegmentator mask instead? (For anatomy-informed prompts)
> >
> > TotalSegmentator requires predefined labels. FLanS enhances workflow efficiency by reducing the steps between a clinician's assessment and the segmentation output. Regarding the way of asking ChatGPT, that is essentially using the Large Language Model's language understanding ability, our proposed method provides an end-to-end way to perform the similar task.
> >
> > > Q2: How did you account for the "Segment the leftmost organ" not changing when iterating along 2D slices? -- How would you imagine this prompt should be applied in practice?
> >
> > The model actually interprets each slice independently by analyzing the semantic features of the organ within that specific 2D context, enabling it to adjust to variations in organ positions or appearances. Besides, our training process assigns labels dynamically (as in Fig.3) based on metrics like area or spatial positioning, ensuring the model learns to recognize and distinguish organs accurately, even if their spatial relations shift across slices.
> >
> > > Q3: Did you evaluate fully in 3D or did you only consider a 2D slice by itself?
> >
> > In the paper, we currently take the 2D slices of the medical image to perform the inference and prompt segmentations.

---

> ### Comment · Reviewer_7cgt · 2024-11-25
> **Re Response**
>
> Thanks for your response to the raised points.
>
> > R1: We propose to handle free form text prompt in two aspects: anatomy-informed and anatomy-agnostic, as for this question, anatomy-informed is indeed to handle those scenarios that users can explicitly describe what organ they expect to segment, which is the simplest case, _we generalize the use case to the diagnosis report that the complex clinical descriptions may not be reflecting specific organ name directly, such as "hydronephrosis" may refer to the "kidney" in the report._
>
> I am aware that this is the proposed capability but the motivation for this is still in question, mirroring concerns of yQbg.
> Why would a clinician not just create another prompt? Why does one need to infer the target organ simplicitly? This needs to be  highlighted or debated more. Currently, this "clinically relevant" statement is made but no reasoning for the use-cases are given.
>
> > R2: [...] Our current 2D approach is an initial step towards more complex, fully integrated 3D segmentation capabilities.
>
> This is currently not clearly stated and should be denoted more prominently, e.g. in the Introduction and always in the limitations section. Overall though I agree that this may be a desirable capability, the current experiments are just not there yet to show this. Personals e.g. some conditional segmentation would already be sufficient. E.g. "segment all lesions that are on the boundary of an organ" or switching segmentation instructions to "Segment the liver without hepatic vessels/excluding tumors".
>
>  >R3: Even though the organ segmentation task itself has been an interest with multiple methods, this paper focuses on a free-form text trigger, which offers a more tailored, interactive, and context-aware solution.
>
> Until you provide more convincing experiments than basic organ segmentation this will remain a glaring weak-point of the paper. Inclusion of one of the above proposed experiments could address this though.
>
>  >R4: canonicalization
>
> Fair enough, sometimes data may also not get entered correctly, yet I believe the canonicalization is not of that high importance or interest. Maybe move larger parts to the supplement instead. -- But if you argue about `clinical relevance` one may expect clinics to be able to handle correctly orientating images.
>
> >R5:  Evaluation
>
> Maybe add a sentence to describe this more clearly.
> Also maybe add some information on how exactly the 2D slices were evaluated in the main. The information about exclusion of small organ volume is a limitation that should be discussed and not hidden away.
>
> >R6: Baselines
>
> Universal Models may be a baseline but there can't be enough. Given nnU-Nets prominence you could easily add a 2D nnU-Net. Personally, I would like to see a 3D nnU-Net to just get a reference on how close the model is to the state-of-the-art. I wouldn't mind it being lower in performance but that information and limitation discussed would be great.
> (But if one compares to nnU-Net the full 3D volume needs to be converted to 2D slices. No ignoring of empty slices or slices with low organ volume)
>
> Overall I believe the provided response is insufficient to change my score. No content has been added so far, even though the promptable nature of the model should allow for it rather easily. Hence I maintain my stance, but I hope that the authors consider including some proposed suggestions and resubmit, as I believe the direction to be interesting and promising.

---

### Official Review · Reviewer_Nnys · 2024-11-04

**Soundness:** 3
**Presentation:** 3
**Contribution:** 2
**Rating:** 3
**Confidence:** 4

**Summary:**

This paper aims to explore the capability of text for segmentation in real clinical setting, the main contributions of this paper can be summarized as following:
1) Proposed a retrieval-augmented generation (RAG) based free form text prompt generator to enhance the segmentation performance
2) Proposed a symmetry-aware canonicalization module to ensure same scan orientations and reduce anatomical position
3) Demonstrated effectiveness on both in-domain and out-of-domain datasets

**Strengths:**

The strength of this paper can be summarized as follows:
1) It is interesting to see how the variable of text description affects the segmentation performance
2) Adapting RAG to generate description is an interesting way to tackle the generalization capability using text

**Weaknesses:**

The weakness of paper can be summarized as follows:
1) Lack of experiments of comparing with 2D current state-of-the-art supervised model (i.e. nn-UNet, Swin-UNet)
2) The image orientation to train a model cannot be counted as a novel problem to handle, as it is not a problem and it is just a correction to the experiment setting.

**Questions:**

1) When we perform training with medical images, changing the orientation to a consistent orientation (i.e. RAS) is a usual step to preprcoess the data. It will be great if you can clarify why you think it is a clinical problem and needs a machine learning model to correct the orientation?

2) From Figure 3, you are using point / id to identify the organ semantic in the image and want to use the generator to correspond each organ with text, which is an interesting idea. However, how about some small organs in the abdominal region? It is challenging to describe all the organ location is, as they are nearby and is there any experimental setting that you have tried to adapt all organs?

3) Similar work have been done by adapting text into segmentation, please cite this paper in the related work and it will be great to see if there is an experiment comparison:
- Zhao, Theodore, et al. "BiomedParse: a biomedical foundation model for image parsing of everything everywhere all at once." arXiv preprint arXiv:2405.12971 (2024).

4) For Figure 7, the t-SNE plot for text-prompt embedding should be completely separable, because the text is pointing towards different organs. A meaningful visualization towards your model should be an attention map of computing the correlation between text and the image. It will be great to have the attention map that demonstrates the shape of the corresponding organs and show the correlated semantics between text and our segmentation target.

---

> ### Author Response · Authors · 2024-11-22
> **Replies to the reviwer's concerns**
>
> We really appreciate it for the reviewer's time and valuable comments. We will address the concerns as follows:
>
> ### Weakness
>
> > Lack of experiments of comparing with 2D current state-of-the-art supervised model (i.e. nn-UNet, Swin-UNet)
>
> 1. MedSAM, included in our baseline comparisons, has been demonstrated to outperform many of the Unet-based methods [1]. By outperforming MedSAM, FLanS indirectly demonstrates its superiority over these U-net-based models, illustrating advanced capabilities across various benchmarks.
>
> 2. The nn-UNet and Swin-UNet are optimized for traditional segmentation tasks and excel when provided with fixed labels or point annotations. However, they are not designed to handle the diverse free-form text prompts that are supported by FLanS. Our model's primary contribution is its ability to segment medical images based on natural language commands, which is beyond the capabilities of nn-UNet and Swin-UNet.
>
> We have included the discussions regarding this in our paper's updated version of related work.
>
> [1] Ma J, He Y, Li F, et al. Segment anything in medical images[J]. Nature Communications, 2024, 15(1): 654.
>
> > The image orientation concerns regarding the novelty
>
> 1. By incorporating the symmetry-aware canonicalization module, FLanS achieves inherent invariance to changes in image orientation. This means that regardless of how the input image is oriented, the model’s performance remains consistent and reliable. This invariance is crucial for deploying medical image analysis systems in real-world settings, where such variations are common.
>
> 2. FLanS’s canonicalization network is not just a preprocessing step but a dynamic component that optimizes image orientation to enhance segmentation accuracy. It actively learns and adjusts to present images in the most favorable orientation for the segmentation network, significantly boosting its performance.
>
> 3. As shown in Fig. 1 of our paper, CT images from different datasets can exhibit significant variations in orientation, posing a substantial challenge for medical applications. The canonicalization network in FLANs offers an automated solution for detecting and aligning image orientations, even in the absence of raw DICOM data. This flexibility makes it particularly suitable for privacy-sensitive medical applications, such as segmentation.
>
> We’d like to point out that, the `Symmetry-Aware Canonicalization Network is fundamentally different from data augmentation`, while augmentation introduces artificial variability during training, canonicalization dynamically aligns all images to a consistent, task-relevant orientation before segmentation. By incorporating symmetry awareness in a learned network, it resolves orientation ambiguities (e.g., left-right flips in symmetrical organs such as kidneys) that augmentation cannot address, which is crucial for free-form text-based segmentation.
>
> We also have experiments demonstrated (as shown in Table. 4) that segmentation accuracy improves significantly with canonicalization, even when compared to models trained with heavy augmentation processing.
>
> ### Question
>
> > When we perform training with medical images, changing the orientation to a consistent orientation (i.e. RAS) is a usual step to preprcoess the data. It will be great if you can clarify why you think it is a clinical problem and needs a machine learning model to correct the orientation?
>
> 1. we observed significant orientation discrepancies across different datasets.
>
> 2. FLanS is designed as a foundation model for medical image segmentation. A foundation model must perform consistently well under a wide range of clinical imaging conditions and not be limited to a predefined orientation like RAS.
>
> 3. Re-orientation can be challenging, as the required operations often vary across datasets and typically rely on scan header information, which may not be available in public datasets, regardless of the preprocessing tool used.
>
> 4. By integrating orientation optimization directly within the model’s learning framework, FLanS achieves a higher degree of robustness and accuracy. This approach is particularly valuable in clinical environments where data variability is the norm, not the exception.

---

> ### Author Response · Authors · 2024-11-22
> **Cont. of the replies**
>
> ### Question
>
> > From Figure 3, you are using point / id to identify the organ semantic in the image and want to use the generator to correspond each organ with text, which is an interesting idea. However, how about some small organs in the abdominal region? It is challenging to describe all the organ location is, as they are nearby and is there any experimental setting that you have tried to adapt to all organs?
>
> In our approach, we utilize bounding boxes to label organ locations and sizes. For smaller organs, we distinguish them based on the area of their bounding boxes and incorporate this information into the text prompts used for training. These prompts explicitly describe positional and size-related details, enabling the model to better understand and differentiate smaller organs.
>
> We acknowledge the challenge of precisely describing the positions of all organs, especially in regions where organs are closely packed. `However, our method leverages the semantic alignment capability of the text encoder. By grounding positional semantics (e.g., left, right, top, bottom) through identifiable major organs, the model learns a positional comprehension framework that extends to smaller or less distinct organs`. This integration provides a structured way for the model to infer broader spatial relationships and improve its understanding of the entire anatomical context.
> So in our work, even though describing every organ's position is inherently difficult, our approach balances detailed labeling for small organs with a learned positional framework, enabling the model to achieve robust performance across various organ types.
>
> > Similar work have been done by adapting text into segmentation, please cite this paper in the related work and it will be great to see if there is an experiment comparison.
>
> Thanks a lot for the suggestion! This is a very valuable paper and working on similar tasks.
>
> We have incorporated the citation and discussion of this paper. However, we noticed that the referred paper just released the code today, so we are working on integrating the experiment for comparison, and we will surely provide empirical results if time allows.
>
>
> > For Figure 7, the t-SNE plot for text-prompt embedding should be completely separable, because the text is pointing towards different organs. A meaningful visualization towards your model should be an attention map of computing the correlation between text and the image. It will be great to have the attention map that demonstrates the shape of the corresponding organs and show the correlated semantics between text and our segmentation target.
>
> 1. Thanks for the suggestion, we have included the attention map calculated by the dot product of the two embeddings (text and the image), as shown in the Appendix.
>
> 2. In the appendix, for example, consider the attention map generated with the prompt `...segment the part located at the topmost portion…(as highlighted by the red box)`; it does not highlight just one organ, instead, all organs at the top are highlighted. This demonstrates that our model is not merely overfitting data to infer a specific organ; rather, it has a deep understanding of the text and its relationship to the medical image.
>
>
>
> **We really appreciate all of your suggestions, especially for the `attention map` that really helps us to better illustrate the bind of model's text and image embeddings, and we will provide wider discussions to solve the potential unclearness in the paper**.

---

### Note · Authors · 2024-12-05

I have read and agree with the venue's withdrawal policy on behalf of myself and my co-authors.